# A Novel Non-Invasive Murine Model of Neonatal Hypoxic-Ischemic Encephalopathy Demonstrates Developmental Delay and Motor Deficits with Activation of Inflammatory Pathways in Monocytes

**DOI:** 10.3390/cells13181551

**Published:** 2024-09-14

**Authors:** Elise A. Lemanski, Bailey A. Collins, Andrew T. Ebenezer, Sudha Anilkumar, Victoria A. Langdon, Qi Zheng, Shanshan Ding, Karl Royden Franke, Jaclyn M. Schwarz, Elizabeth C. Wright-Jin

**Affiliations:** 1Division of Biomedical Research, Nemours Children’s Health, Wilmington, DE 19803, USA; elise.lemanski@nemours.org (E.A.L.); bailey.collins@nemours.org (B.A.C.); andrew.ebenezer@nemours.org (A.T.E.); sudha.anilkumar@nemours.org (S.A.); victoria.langdon@nemours.org (V.A.L.); qi.zheng@nemours.org (Q.Z.); karl.franke@nemours.org (K.R.F.); 2Psychological and Brain Sciences, University of Delaware, Newark, DE 19716, USA; jschwarz@udel.edu; 3Applied Economics and Statistics, Center for Bioinformatics and Computational Biology, University of Delaware, Newark, DE 19716, USA; sding@udel.edu; 4Sidney Kimmel Medical College, Thomas Jefferson University, Philadelphia, PA 19107, USA; 5Division of Neurology, Nemours Children’s Health, Wilmington, DE 19803, USA

**Keywords:** hypoxic-ischemic encephalopathy, maternal immune activation, motor, development, microglia, macrophages

## Abstract

Neonatal hypoxic-ischemic encephalopathy (HIE) occurs in 1.5 per 1000 live births, leaving affected children with long-term motor and cognitive deficits. Few animal models of HIE incorporate maternal immune activation (MIA) despite the significant risk MIA poses to HIE incidence and diagnosis. Our non-invasive model of HIE pairs late gestation MIA with postnatal hypoxia. HIE pups exhibited a trend toward smaller overall brain size and delays in the ontogeny of several developmental milestones. In adulthood, HIE animals had reduced strength and gait deficits, but no difference in speed. Surprisingly, HIE animals performed better on the rotarod, an assessment of motor coordination. There was significant upregulation of inflammatory genes in microglia 24 h after hypoxia. Single-cell RNA sequencing (scRNAseq) revealed two microglia subclusters of interest following HIE. Pseudobulk analysis revealed increased microglia motility gene expression and upregulation of epigenetic machinery and neurodevelopmental genes in macrophages following HIE. No sex differences were found in any measures. These results support a two-hit noninvasive model pairing MIA and hypoxia as a model for HIE in humans. This model results in a milder phenotype compared to established HIE models; however, HIE is a clinically heterogeneous injury resulting in a variety of outcomes in humans. The pathways identified in our model of HIE may reveal novel targets for therapy for neonates with HIE.

## 1. Introduction

HIE is a common brain injury that affects infants born at term with an estimated incidence of 1–3 per 1000 births in developed countries and 26–30.6 per 1000 births in underdeveloped countries [1,2]. HIE can be caused by a myriad of birth complications including placental abruption, uterine rupture, cord prolapse, chorioamnionitis, and maternal hypotension. Ultimately, these factors lead to insufficient delivery of oxygen to the fetal brain, resulting in the risk of permanent brain injury. Children with severe HIE have a mortality rate of up to 50% and those who survive can have significant long-term neurologic deficits including cerebral palsy, epilepsy, and vision/hearing impairments [3,4]. Therapeutic hypothermia is the only effective therapy for HIE and involves cooling the infant to a temperature of 33.5 °C for 72 h [5]. Despite the success of this therapy in improving outcomes, up to 40% of neonates who receive this treatment still suffer brain injury and disability [5,6]. To be effective, therapeutic hypothermia must be initiated in the first 6 h of life. If clinical signs of hypoxia are missed, the infant can quickly move out of this narrow window for treatment [7]. This demonstrates the critical need for additional neurotherapeutics to mitigate brain injury and reduce lifelong disabilities following neonatal HIE.

One of the most prominent risk factors for HIE is inflammation. Forty to fifty percent of neonates affected by neonatal HIE are born to mothers with chorioamnionitis or clinical signs of this infection, such as fever and leukocytosis [8]. However, most preclinical models of HIE do not take inflammation into account [9]. MIA is a well-established model of inflammation that uses a peripheral injection of lipopolysaccharide (LPS) during gestation to elicit an immune response [10]. LPS is a bacterial cell-wall-derived endotoxin that binds to toll-like receptor 4 (TLR4) on immune cells to stimulate the production of immune molecules, including cytokines, in the dam [11]. This immune response is also triggered in the fetus by the transmission and production of cytokines through the placenta [10,12]. Although maternal infection is a known risk factor for developmental delays and disorders in humans, most cases of maternal infection or inflammation do not lead to these outcomes [13]. Therefore, it is hypothesized that maternal immune activation may have a priming effect that leads to increased susceptibility to environmental or genetic “second hits”, further increasing the risk of developing various CNS disorders [14,15] depending on the timing, severity, and type of cumulative stressors [16].

Our novel model of neonatal HIE, characterized here, combines MIA via systemic LPS injection in late gestation with a short but severe global hypoxia at postnatal day 6 (P6). Exposure to LPS during late pregnancy effectively simulates chorioamnionitis and is used in other models of this common pregnancy complication [17]. Utilizing MIA prior to hypoxia allows for the investigation of the interaction of etiologically relevant maternal, placental, and neonatal inflammatory factors that contribute to the complex brain injury of HIE around the time of birth. We excluded the carotid artery ligation used in the popular Rice–Vannucci model due to the substantial hemispheric ischemic damage that this model evokes. We sought to thoroughly characterize our model of HIE to investigate whether it recapitulates the outcomes and symptomatology present in humans. This characterization includes investigating the changes within microglia following injury, gross changes in brain volume 24 h following the second hit of hypoxia, determining whether this model leads to developmental delays in the neonatal period, and determining long-term deficits in motor and social function. These results help validate the proposed model and inform metrics to examine treatments tested within this model.

## 2. Materials and Methods

### 2.1. Mouse Strains

CF-1 mice were acquired from Charles River as timed pregnant dams. Mice had a 12 h light/dark cycle with free access to food and water. Determination of sex in neonatal mice less than 10 days old was achieved by genotyping for *Sry* using the following primers.

SRY F: TTG TCT AGA GAG CAT GGA GGG CCA TGT CAA.

SRY R: CCA CTC CTC TGT GAC ACT TTA GCC CTC CGA.

### 2.2. Maternal Immune Activation

Timed pregnant mice were injected with 50 micrograms/kg body weight lipopolysaccharide (LPS) or 0.05 mL 0.9% sterile saline via intraperitoneal injection on embryonic day 18 (E18). In mice, gestation typically lasts an average of 20 days, and E18 is equivalent to the third trimester in humans [18]. Dosing of LPS was determined via survival analysis of dams and litters with poor survival at higher doses.

### 2.3. Hypoxia Exposure

Adapted from [19]. P6 was selected as the time point for hypoxia administration due to poor animal survival at P7 and later time points. This time point was also selected due to its neurodevelopmental equivalency to late preterm gestation, as P7–P10 is considered the murine equivalent of term gestation in humans [20]. Mice were placed in a hypoxia chamber (Biospheryx, Parish, NY, USA) on a heated pad (37.2 °C) and subjected to 8 min of either progressive hypoxia from 21% to 0% oxygen or normoxia (21% oxygen) (Figure 1). After 8 min of hypoxia, the chamber door was opened to allow rapid recovery to 21% oxygen. Surviving mice were returned to their mother for further recovery. This procedure had a mortality rate of 7% (*n* = 43).

### 2.4. Neonatal Development Testing

On the day of birth (P0), the number of offspring was counted, and the pups were weighed. Litters were culled to a maximum of ten pups, and litters of less than six pups were excluded, to control for differences in behavior that could be attributed to litter size induced by the abortifacient effects of LPS (Appendix A). Beginning on P1, between two and four males and females from each litter were examined daily for the acquisition of typical developmental milestones and reflexes. Testing was performed at the same time each day. The pups from each litter were removed from the dam and kept on a heating pad at 37 °C to maintain a stable body temperature during testing. The means from the males and females from each litter were used for statistical comparisons to avoid litter effects. Testing for each reflex began three days prior to the typical onset of the behavior, when possible, and was performed until the response was observed for two consecutive days. Behavioral tests were adapted from [21] including eye opening, surface righting, negative geotaxis, rooting, forelimb grasp, auditory startle, open area traversal, and air righting. Hindlimb splay, an assessment of gross motor function and muscle tone, was additionally included. For this test, beginning on P5, each pup was suspended from its tail, and hind limb extension was observed. When the pup fully extended both hindlimbs to 45 degrees, this was recorded as the acquisition of hindlimb splay.

### 2.5. Adult Behavior

The following behavioral assessments were performed on adult mice beginning at P60.

Grip Strength. Mice were lowered to grab a triangular pull bar with their forelimbs on a grip strength meter (Columbus Instruments, Columbus, OH, USA) and were pulled backward by the tail until they lost their grip. Force in newtons applied to the bar before release was recorded across two trials and averaged.

Catwalk. Each mouse was placed on the platform on the CatWalk (Noldus, Leesburg, VA, USA) and the gait pattern of each mouse was captured videometrically and subsequently analyzed using the software package for the apparatus, Catwalk XT (Noldus, version 10.6). Three compliant trials with criteria of a minimum run duration of 0.5 s and a maximum run duration of 5 s were recorded per animal. Trials were additionally excluded if they did not meet the criteria of a minimum number of 10 consecutive steps per run, an average speed range from 30 to 90 cm/second, and a maximum speed variation of 40%, or if the animal stopped during the trial.

Rotarod. Mice were habituated to the apparatus in two 2 min sessions 2–3 h apart the day prior to testing at a constant speed of 4 rpm. On the day of testing, each mouse was placed onto a moving drum of a Rotarod Treadmill for Mice (Ugo Basile, Gemonio, Italy). The rotarod treadmill was set to accelerate progressively from 4 to 40 rpm over 300 s. The amount of time the mouse remained moving on the drum was recorded. Three trials were performed with a 15 min inter-trial interval.

Sociability. Mice were placed in a plexiglass box 48″ in length and 16″ in height separated into three smaller chambers. On day 1, the mouse was habituated to the apparatus. The mouse was placed in the middle chamber and left to explore the apparatus for 5 min. The mouse was then removed, and an object was added to the cage in a lateral chamber, and a species-, sex-, and age-matched novel mouse was added to the cage in the opposite chamber. The experimental mouse was then placed into the middle chamber and allowed to explore for 5 min. Sessions were video recorded, and the amount of time spent with the novel animal or the novel object was measured. Day 2 consisted of a social memory test. One chamber contained the familiar mouse from Day 1 and the other a new novel mouse. The experimental mouse was then placed into the center chamber and allowed to explore the apparatus for 5 min. Sessions were video recorded, and time spent with the novel or the familiar mouse was measured.

### 2.6. Structural MRI

Brains from a subset of offspring were collected on P7 for ex vivo MRI, 24 h after hypoxia. Animals exposed to maternal immune activation alone were included in this assessment to control for weight differences as animals born to LPS-exposed mothers weighed less than controls (Appendix A). MRI was performed with a Bruker Biospec 94/20 (Billerica, MA, USA) with a 9.4 Tesla magnet at the University of Delaware Center for Biomedical and Brain Imaging (CBBI). A T2-weighted 2D structural MRI was conducted on brains collected at P7. Manual segmentation was performed on structural scans using ITK-SNAP (v4.0.1) to determine the relative size of the hippocampus, dorsal striatum, and cortex. In adult brains, a T2-weighted 3D structural scan was conducted to identify if structural differences persist into adulthood. Structural MRI scans were manually segmented for regions of interest (ROIs).

### 2.7. Bulk RNAseq

Sequencing. Whole brains were collected from male and female pups one day following the second hit of hypoxia (P7) and dissociated into a single-cell solution using the Adult Brain Dissociation Kit and gentleMACS Octo Dissociator (Miltenyi Biotec, Gaithersburg, MD, USA). Microglia were enriched via magnetic CD11b-coated bead isolation (Miltenyi Biotec, Geithersburg, MD, USA). RNA was extracted and prepared by the Pediatric Genomics Laboratory at Nemours Children’s Health using the Illumina Stranded Total RNA Prep with Ribo Zero Plus. RNA sequencing was performed via Illumina NextSeq 500/550 High Output Kit v2.5 (300 cycles) conducted by the Pediatric Genomics Laboratory at Nemours Children’s Health (Wilmington, DE, USA).

Bioinformatic Analysis. Bulk RNAseq libraries were mapped to the GRCm39 genome assembly using Sentieon’s [22] accelerated version of the STAR [23] v2.7.10b algorithm. Gene counting was performed via RSEM [24] v1.3.1. Differential gene expression analysis was performed using edgeR (v3.40.2) [25] and DESeq2 (v0.38.3) [26] by taking a union of the results.

Gene set enrichment analysis. GSEA was performed using the GSEA and MSigDB software (v4.3.3) available as a joint project of UC San Diego and the Broad Institute [27]. GSEA Preranked analysis was performed using mouse hallmark gene sets [28] and the preranked expression from DEG log2FC results with an FDR-adjusted *p*-value < 0.05. Cutoffs of the nominal *p*-value of <0.001 and FDR < 0.05 were used for the inclusion of statistically relevant gene sets.

### 2.8. scRNAseq

Whole brains were collected from male and female pups two days (P8) and four days (P10) day following the second hit of hypoxia. ScRNAseq was performed by the creation of a single-cell solution using the GentleMACS brain dissociator (Miltenyi Biotec, Gaithersburg, MD, USA) and Adult Brain Dissociation Kit (Miltenyi Biotec, Gaithersburg, MD, USA). Cell viability was confirmed via trypan blue staining. Live and dead cells were counted using a hemocytometer at 10× magnification. The 10× Genomics Chromium Next GEM Single Cell 3′ Kit v3.1, Dual Index Kit TT Set A, Chromium Next GEM Chip G Single Cell Kit, and SPRIselect Reagent Kit were used for library creation. Sample and library quality control was achieved using Agilent High Sensitivity D5000 ScreenTape, D5000 Reagents, D5000 Ladder, and KAPA Universal Library Quantification Kit. Sequencing was achieved using Illumina NextSeq 2000 P3 Reagents (100 cycles). All scRNAseq libraries were sequenced on an Illumina NextSeq 2000 instrument (Illumina Inc, San Diego, CA, USA) at the Nemours Research Lab with a 2 × 150 paired-end (PE) read setting. Raw FASTQ read files were called using the Illumina Dragen software (v4.2.7).

All scRNAseq libraries were sequenced on an Illumina NextSeq 2000 instrument at the Nemours Research Lab with a 2 × 150 paired-end (PE) read setting. Raw FASTQ read files were called using the Illumina NextSeq 1000/2000 Control Software (v1.5.0).

### 2.9. scRNAseq Data Processing and Statistical Analysis

The scRNAseq dataset was processed using the 10Xgenomics Cell Ranger “count” pipeline (v7.1.0) designed for the 3′ Gene Expression analysis. In essence, this pipeline first generates barcode-embedded FASTQ files (“mkfastq”), then calculates single-cell level feature/barcode count matrix (“count”) for each sample, for which the Cell Ranger pre-built mm10-2020-A database (GENCODE vM23/Ensembl98) was used. All the scRNAseq count data were imported into an RStudio server at Nemours as R objects using the Seurat package (v5.1.0) [29,30], and subsequent statistical and visualization analyses were performed on the RStudio server using R packages such as Seurat (v5.1.0), DESeq2 (v1.34.0), and gprofiler2 (v0.2.3) [26,29,30,31]. To be specific, Seurat objects for all samples were merged into a single integrated object using the Seurat v5 integration procedure, and cell clusters were identified using the shared nearest neighbor (SNN) method based on the integrated data. To annotate the cell clusters, we downloaded known gene markers of potential cell types from brain tissues from the PanglaoDB and CellMarker 2.0 databases [32,33] (Appendix A), then cross-examined the expression profiles of the marker genes in the cell clusters, so we can manually annotate Seurat identified cell clusters. Notably, a few cell clusters failed to be annotated using the known markers. For this unknown cluster, we identified conserved markers using the “FindConservedMarkers” function from Seurat and manually annotated them as endothelial cells based on prior knowledge of function associated with these gene markers. To identify subclusters for given cell types including microglia, macrophages, and T/B cells, we subset the original integrated dataset based on the cell type annotations, then performed sub-cluster analysis for each aforementioned cell type similarly as for the overall dataset. To identify differentially expressed genes (DE-genes) between different treatment conditions, we generated “pseudobulk” RNAseq objects from Seurat, where single-cell expression profiles from the same sample were aggregated, and further converted them into pseudobulk RNAseq datasets using the DESeq2 package. DE genes of various comparisons were identified by negative binomial models using the DESeq2 package [26]. To be specific, we used the following negative binomial model of “Count ~ Timepoint + Sex + Celltype + Treatment + Celltype:Treatment”, in which the individual terms were stepwise optimized with repeated Likelihood ratio tests using the nbinomLRT function from the DESeq2 package. Finally, the functional and pathway enrichment analysis was performed using the gost function from the gprofiler2 package [31].

### 2.10. Statistics

MRI. Statistical analyses were performed using GraphPad Prism (version 10.2.0). Two-way ANOVAs were used for analysis with HIE and sex as factors. When MIA was considered as a factor, a two-way ANOVA was initially performed and when no effect of sex was found, a one-way ANOVA was performed with Controls, MIA, and HIE as factor levels.

Behavior. Statistical analyses were performed using GraphPad Prism (version 10.2.0). Comparisons were made with two-way ANOVA using sex and HIE treatment as factors. For the acquisition of neonatal behaviors, males and females were averaged for each litter for a total of two data points per litter, as litter effects are particularly prominent in the neonatal period [21]. The data from individual animals were analyzed for adult behavior to capture the full variance of adult behavior. Data are shown as mean ± SEM, with individual data points included on graphs. *p* < 0.05 was considered statistically significant.

Rotarod. Statistical analysis was conducted on R (version 4.3.3). Data were not normally distributed due to a cutoff of 300 s. To take both latency and censorship into account across trials a Cox mixed-effects model *“coxme()”* was used. The model incorporated fixed effects of treatment and sex, random effects of trial and trial/treatment, and a nested factor of 1. We removed sex as a factor when no significant main effect or interaction with sex was found. A *p* value less than 0.05 was considered statistically significant.

## 3. Results

### 3.1. Non-Invasive Two-Hit Model of Neonatal HIE Produces Developmental Delays and Reduction in Brain Volume

Overall whole-brain volume did not show a significant difference on ex vivo MRI analysis at P7 (Figure 1A) (*F* (1, 12) = 0.0612, *p* = 0.0842, two-way ANOVA). However, when animals who received the single hit of MIA were included to help control for decreased weight due to MIA, there was a trend toward significance that appears to be driven by the two-hit HIE exposure (Appendix A, *F* (2, 18) = 3.270, *p* = 0.0615, one-way ANOVA). There were no significant differences in brain region volume when controlling for whole-brain size (Appendix A).

HIE animals had a significant delay in acquisition in five of the nine behavioral assessments (Statistics summarized in Table 1). There was a significant main effect in delay of acquisition for HIE animals compared to control animals in the following behaviors: rooting (Figure 2B), negative geotaxis (Figure 2C), hindlimb splay (Figure 2F), open area (Figure 2G), and air righting (Figure 2H). There were no differences in righting (Figure 2H), forelimb grasp (Figure 2E), auditory startle (Figure 2I), or eye opening (Figure 2J).

### 3.2. Non-Invasive Two-Hit Model of HIE Results in Adult Motor Deficits in Gait and Grip Strength

In the catwalk assessment, stride length, swing, and speed were chosen a priori as parameters for analysis. HIE animals exhibited shorter stride length in both forepaws (Figure 3A). They also exhibited a shorter swing in hindpaws (Figure 3B), but not forepaws (Figure 3B). Despite this difference in gait, they did not have any difference in overall body speed compared to controls (Figure 3C). HIE animals had a weaker grip strength compared to controls (Figure 3D), and females had a weaker grip strength compared to males. There was no interaction between HIE and sex. Catwalk and grip strength statistics are summarized in Table 2. Rotarod is a test of motor coordination and motor learning. HIE animals stayed on the rotarod significantly longer compared to controls (Figure 3E, Table 3). There were additionally no differences in either the three-chamber sociability test or the three-chamber social novelty test. (Appendix A).

### 3.3. Non-Invasive Two-Hit Model of HIE Produces Immediate Inflammatory Changes in Microglia

Microglia (CD11b+) cells were isolated from whole brains for 24 h following hypoxia on P7. A total of 1335 genes were found to be differentially expressed with an FDR-adjusted *p*-value of less than 0.05 by both edgeR and DESeq2. Of those, 157 were upregulated and 1178 were downregulated (Figure 4). A gene set enrichment analysis (GSEA) of mouse hallmark genes preranked by DEseq2 identified 15 significantly upregulated gene sets (FDR *q*-value < 0.05) (Table 4) and 4 downregulated gene sets (Appendix A). Several of the upregulated gene sets in microglia 24 h following hypoxia (P7) represent a classical proinflammatory profile within microglia (TNFα via NFκB, Interferon -α and -γ responses, IL6/JAK/STAT3 Signaling, Inflammatory Response, Complement, IL2/STAT5 Signaling, Figure 4B). Other gene sets represent the upregulation of cellular proliferation (MYC Targets V1, MYC Targets V2, PI3K/AKT/mTOR Signaling, and E2F targets, Figure 4C), as well as DNA damage checkpoint (G2M checkpoint), and apoptosis (Figure 4D). A GSEA was performed on bulk RNAseq results from microglia isolated from whole brains collected 8 days following hypoxia (P14), and 215 genes were found to be differentially expressed with an FDR < 0.05 in both edgeR and DESeq2 (Appendix A). No gene sets were significantly different between groups at this time point. No sex differences were found (Appendix A).

### 3.4. scRNAseq Reveals Monocyte Subclusters of Interest in HIE

Thirteen microglia and five macrophage subclusters were identified on P8 and P10 (Figure 5B,C). No novel subclusters were observed in HIE vs. control animals in either cell type. Analysis for differentially expressed genes in microglia with significant HIE and subcluster interactions found 27 upregulated genes (Table 5a) and 23 downregulated genes (Table 5b). Genes with distinct subcluster locations were primarily in subclusters 7, 11, and 12, indicating that these may be subclusters of interest in the microglia response to HIE. Pathway enrichment analysis of these subclusters’ significantly over-expressed marker genes shows that subcluster 7 (Figure 6A) and subcluster 12 (Figure 6B) are enriched for neuron and nervous system development pathways. Subcluster 11 only had two significant over-expressed marker genes, *Hba-a1* and *Hbb-bs*, both of which are hemoglobin genes (Appendix A).

### 3.5. scRNAseq Reveals Changes in Microglia Motility, Macrophage Regulation of Neuron Development, and Epigenetic Pathway Upregulation in Macrophages after HIE

Pseudobulk RNAseq analysis of microglia and macrophages was performed using the scRNAseq data from brains collected on P8 and P10. In microglia, 125 genes were upregulated and 5 were downregulated (Figure 7A), and pathway enrichment analysis revealed 218 significantly different pathways (Figure 7B). ReviGO analysis of the upregulated pathways demonstrates that microglia have significantly upregulated genes that are primarily involved in the actin cytoskeleton and cell motility (Appendix A). These include the biological processes of negative regulation of supramolecular fibers, actin polymerization, chemotaxis, cell localization, the molecular function of cytoskeletal protein binding, and the cellular components of anchoring junction, plasma membrane protein complex, lamellipodium, and actin cytoskeleton. There is also significant upregulation in genes involved in neuron development and synaptic signaling. In macrophages, 24 genes were upregulated, and 6 genes were downregulated (Figure 8A). Pathway enrichment analysis revealed 237 significantly different pathways (Figure 8B). ReviGO analysis of macrophages revealed biological processes primarily involved in neuronal and brain development (Appendix A). Changes in molecular function and cellular component genes point to upregulation of epigenetic changes specifically in the macrophage population (Appendix A). Interestingly, epigenetic changes were not seen in the microglia population.

## 4. Discussion

This novel two-hit mouse model of neonatal HIE uniquely allows for the investigation of maternal risk factors in the pathogenesis of HIE. Specifically, maternal infection and inflammation are optimally investigated in this system. This model also poses advantages over the traditional Rice–Vannucci model because it is non-invasive and does not require exposure to anesthesia, which raises concern for added neurotoxicity [34]. Our model utilizes maternal immune activation on E18, considered equivalent to the third trimester of pregnancy in humans [35], and a short severe global hypoxia on P6, considered equivalent to late preterm in human infants [20]. The two-hit model of neurodevelopmental disorders suggests that early life adversity such as MIA leads to increased risk in combination with a later stressor [36]. MIA has been shown to lead to alterations in immune response following a second immune challenge or stressor [14,37], and prior studies have demonstrated that MIA leads to exacerbated immune and autism-like behavioral outcomes when followed by the Rice–Vannucci model of HIE [38]. This is thought to occur through microglia priming, which occurs when microglia become sensitized after entering a proinflammatory state following an initial stressor. However, the two-hit hypothesis of MIA has not been observed in all cases of non-immune second hits [39].

Ex vivo MRI resulted in a trend toward significance with two-hit HIE animals exhibiting a relative decrease in whole brain volume only one day following the second hit of hypoxia. The lack of significance may be due to a higher variability in HIE animals as outcomes following HIE injury are often heterogeneous. MIA-only animals were included in only this measure to control for the effects of LPS on overall pup size. However, the MIA animals had similar brain volumes to controls. These results are similar to those in humans, where infants have demonstrated decreased subcortical brain volumes acutely following injury [40].

HIE animals exhibited delays in the acquisition of developmental reflexes and behaviors predominantly in motor domains, supporting a phenotype comparable to that in human infants after HIE [41,42]. Some of the delays were in skills acquired by controls prior to the second hit of hypoxia on P6, indicating that maternal immune activation alone is responsible for at least some degree of developmental delay. However, the brain MRI volumetric differences indicate that LPS alone is not sufficient to decrease overall brain size, supporting the idea that the two hits are necessary for the full phenotype. This study aimed to describe the full two-hit model of HIE and did not parse the two hits individually. Future studies can further elucidate the individual and combined effects of MIA and hypoxia within this model.

In the catwalk gait analysis, HIE animals had shorter overall stride lengths, and shorter hind paw stride times, indicating that they took shorter and quicker steps when compared with controls. Similar gait disruptions have been observed in other models of HIE [19]. Despite the changes in grip strength and gait, HIE animals performed better on the rotarod. As the rotarod test involves the animals staying on a small rod accelerating in rotation, the specific perturbations in gait observed in these animals may be beneficial to this test. Additionally, the test was censored at 300 s when the rod was no longer accelerating. A longer test or a test at a set speed may assess endurance more directly, which may be impacted in these animals. Despite the rotarod results, the changes in gait and the forelimb weakness observed in grip strength support motor dysfunction in adulthood in HIE animals. In the three-chamber behavioral task, the controls exhibited a high variability in discrimination ratio in both the sociability and social novelty. This indicates an issue with how the animals are performing this task at baseline and makes it difficult to assess any changes due to HIE exposure.

A limitation of our model is the lack of significant interactions between sex and HIE in any of the outcomes. This lack of sex differences is surprising given that MIA and other HIE models often show worse outcomes in males [14,19,43]. In humans, males have traditionally been thought to have higher rates of both morbidity and mortality, as well as higher occurrences of developmental delay and disorders following HIE [44]. However, clinical trials do not often report scores for males separately [43]. Some more recent clinical trials have shown little differences in males and females in control groups or following therapeutic hypothermia treatment [45,46,47]. This suggests that these sex differences may be less significant than originally thought and that treatment may additionally decrease these differences.

The two-hit model of HIE described here does not create the hemispheric, stroke-like focal injury common in the Rice–Vannucci model. While this was the intended goal of this model, it is more difficult to confirm injury presence, severity, and location. The focus of this study was to confirm the motor phenotype within this model throughout the lifespan as well as the acute inflammatory profile following injury. Our findings of long-lasting motor deficits and proinflammation within microglia suggest that neural changes are taking place within our model. This is supported by the relative decrease in overall brain volume at 24 h following hypoxia. Future studies of this model will utilize histological methods to determine the extent and location of injury, as well as investigate white matter disruptions that are typical in HIE injury.

Neuroinflammation is immediately apparent in our model of HIE and can be seen within the first 24 h with an upregulation of multiple proinflammatory pathways including TNF-α signaling via the NF-κB pathway. The upregulation of G2M checkpoint and apoptosis pathways indicate cellular damage and death, while the upregulation of proliferation markers MYC targets (V1 and V2), PI3K/AKT/mTOR signaling, and E2F targets [48,49,50,51] suggest that the undamaged cells have increased proliferation as a mechanism of the proinflammatory response. While some upregulated gene sets are not specifically relevant to microglia (Allograft rejection, Kras Signaling Up), they likely represent an upregulation in proinflammatory and proliferation genes, respectively. This increase in proliferation, inflammation, and cell death occurs only in the acute (24 h) microglia response and returns to baseline one week later (P14). This is an important validation of our model as inflammation following hypoxic-ischemic events is a well-established method of secondary injury, and often the target of therapeutics within preclinical models [52].

ScRNA sequencing revealed 13 microglia cell clusters and 5 macrophage cell clusters within the mouse brain. There were no novel subpopulations that arose within the HIE group; however, differential gene expression within the clusters allowed us to identify three unique subpopulations; cluster 7, cluster 11, and cluster 12, which have a number of differentially expressed genes that are HIE-associated and cluster-associated. Pathway enrichment analysis of the top differentiating gene markers of these subclusters found that both cluster 7 and cluster 12 are largely involved in neurodevelopment, including axon and projection development. As this analysis was performed at a later time point than the initial bulk RNAseq, the microglia may be primarily involved in neuronal repair and promote resiliency following the initial increase in neuroinflammation.

Interestingly, pseudobulk RNAseq analysis identified upregulation of multiple genes involved in epigenetic regulation within the macrophage population, as well as neuron and brain development. Microglia are known to be highly involved in neurodevelopment [53,54]. However, when activated by an immune challenge or otherwise disrupted, microglia may not properly contribute to neuron developmental processes such as synaptic pruning, leading to neurodevelopmental disorders and deficits in cognition, motor function, and sensation [55,56]. The upregulation of these pathways in macrophages may indicate a compensatory mechanism that contributes to repair following injury and resiliency within the mouse brain. At this time point, both the macrophages and microglia may be in a repair state as indicated by the microglia subcluster analysis. Many of the pathways found to be upregulated in microglia pseudobulk analysis are involved in actin cytoskeleton structure and cell motility, indicating that these cells are highly mobile following HIE. Although there was no distinct upregulation in proinflammatory pathways at this time point, the microglia may be responding to chemokine release and moving to areas of injury.

Neonatal HIE leads to life-long impacts in affected infants. While therapeutic hypothermia has been effective in decreasing injury for some children, many children are left with variable degrees of lifelong disability, highlighting the critical need for additional therapies for neonatal HIE. Etiologically relevant models are critical to the development of effective therapies that can be successfully translated to humans. The two-hit model presented here replicates the largest perinatal risk factor for HIE and accurately maintains the maternal–fetal connection while presenting an easier and noninvasive method to induce injury in a murine model. This model also results in long-lasting motor deficits, acute brain volume changes, and a proinflammatory response within microglia, which are representative of the impacts of injury within humans.

## 5. Conclusions

The two-hit HIE model outlined here has several advantages over the standard Rice–Vannucci model, including the procedures’ ease and non-invasiveness. No anesthesia or surgery is required for this model, which may cause added stress or toxicity that is not part of the pathophysiology of disease in humans. The maternal immune activation in this model replicates the maternal, placental, and fetal interactions that are characteristic of HIE in a majority of humans but are often lacking in animal models of HIE. This model results in developmental delays and long-lasting motor changes, acute neuroinflammation, and transcriptional changes in microglia and macrophages. The disadvantage of this model is that the injury is more difficult to localize and produces more mild deficits as compared to the Rice–Vannucci model. The lack of sex differences in our findings is also notable and surprising. Additional and nuanced models of HIE are needed to capture the heterogeneity of injury causes and outcomes. A more complete understanding of the interaction of etiology and cellular and molecular mechanisms within the brain will facilitate the development of new and personalized therapeutics currently limited to a one-size-fits-all treatment.

## Figures and Tables

**Figure 1 cells-13-01551-f001:**
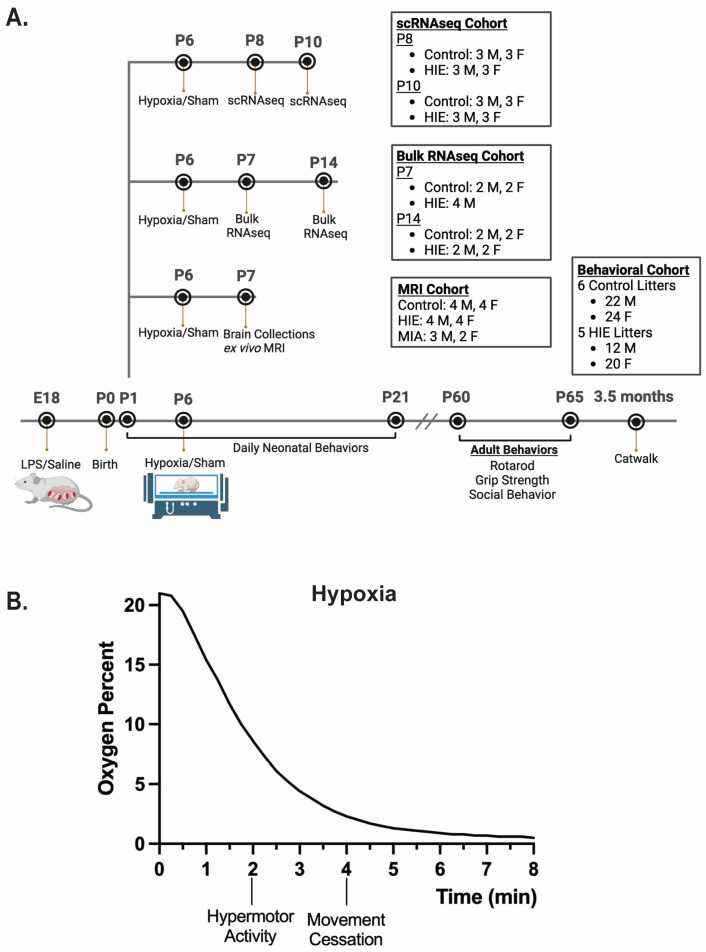
Two-hit HIE model: (**A**) A representation of our two-hit model of HIE and experimental design. (**B**) Representative graph of oxygen levels present and pup behavior during the 8 min hypoxia protocol (*n* = 3 litters).

**Figure 2 cells-13-01551-f002:**
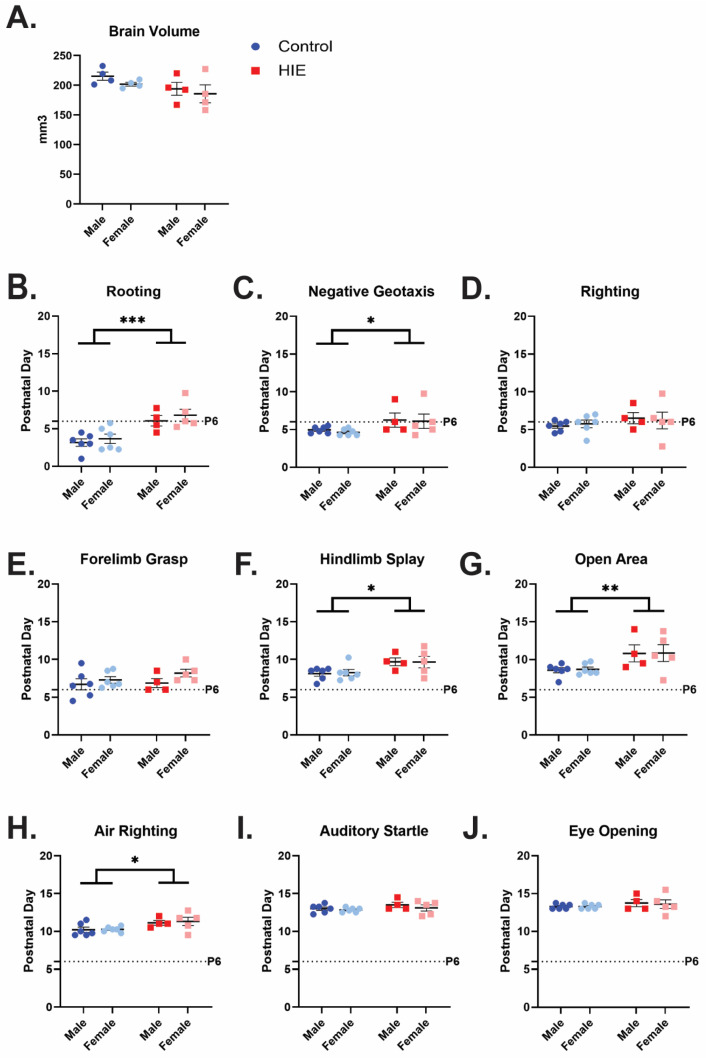
HIE results in a trend toward smaller brains 24 h after injury, and motor developmental delays in the neonatal period: (**A**) Whole-brain volume obtained on P7 through ex vivo MRI for control animals, and two-hit HIE animals. Analyzed with two-way ANOVA (*n* = 4 control male, 4 control female; 4 HIE male, 4 HIE female). (**B**–**J**) Date of acquisition for neonatal developmental behaviors is shown for the average values for males and females in each litter. (*n* = 6 control male, 6 control female; 4 HIE male, 5 HIE female). The dashed line indicates P6, the day of hypoxia exposure. Developmental behaviors were analyzed with a two-way ANOVA. * *p* < 0.05; ** *p* < 0.01, *** *p* < 0.001.

**Figure 3 cells-13-01551-f003:**
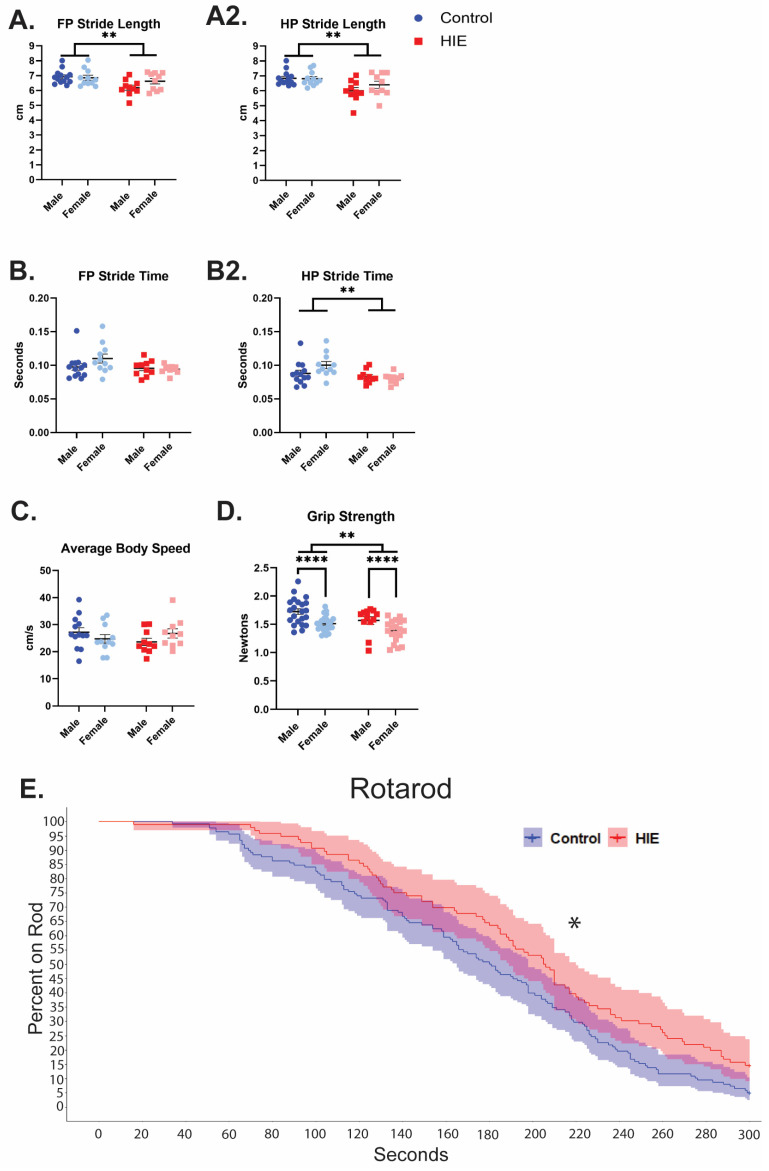
HIE results in distal muscle weakness and gait disturbances in adulthood: (**A**) forepaw (FP) and (**A2**) hindpaw (HP) stride lengths measured by the catwalk ~P105 (two-way ANOVA). (**B**) Forepaw and (**B2**) hindpaw swing time measured by the catwalk (two-way ANOVA). (**C**) Average body speed on the catwalk. (catwalk *n* = 13 control male, 11 control female; HIE = 10 control male, 10 control female, two-way ANOVA) (**D**) Forepaw strength measured by a grip strength meter on ~P60 (*n* = 22 control male, 24 control female; 12 HIE male, 20 HIE female, two-way ANOVA). (**E**) Survival curve showing the proportion of animals still on the rotating rod across time using a Cox mixed-effects model on ~P61. Males and females are collapsed on this graph due to visibility considerations (*n* = 22 control male, 24 control female; 12 HIE male, 20 HIE female, Cox mixed-effects model). * *p* < 0.05, ** *p* < 0.01, **** *p* < 0.0001.

**Figure 4 cells-13-01551-f004:**
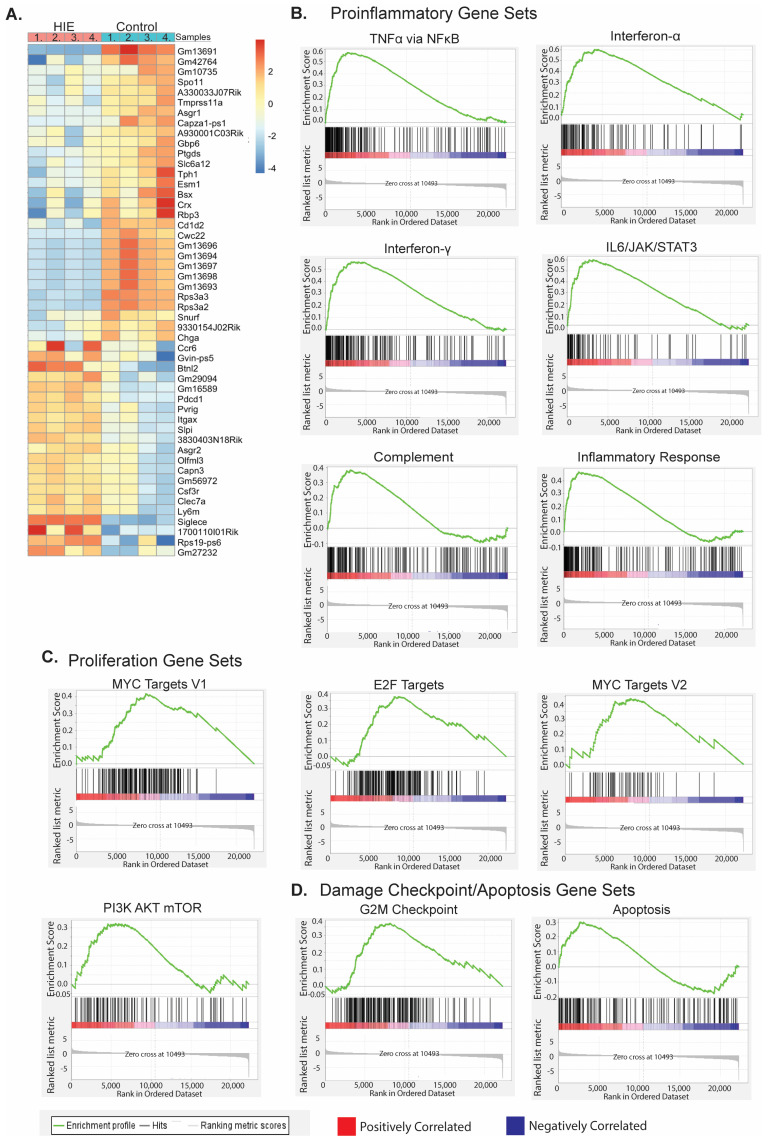
HIE results in acute transcriptional changes within microglia: (**A**) Genes identified by both DESeq2 and edgeR with an FDR adjusted *p*-value < 0.05 within CD11b+ cells on P7, one-day post hypoxia (*n* = 2 control male, 2 control female, 4 HIE male). (**B**) Gene set enrichment plots of significantly upregulated proinflammatory gene sets within HIE microglia. (**C**) Gene set enrichment plots of significantly proliferation-related gene sets within HIE microglia. (**D**) Gene set enrichment plots of significantly upregulated damage checkpoint/apoptosis gene sets within HIE microglia.

**Figure 5 cells-13-01551-f005:**
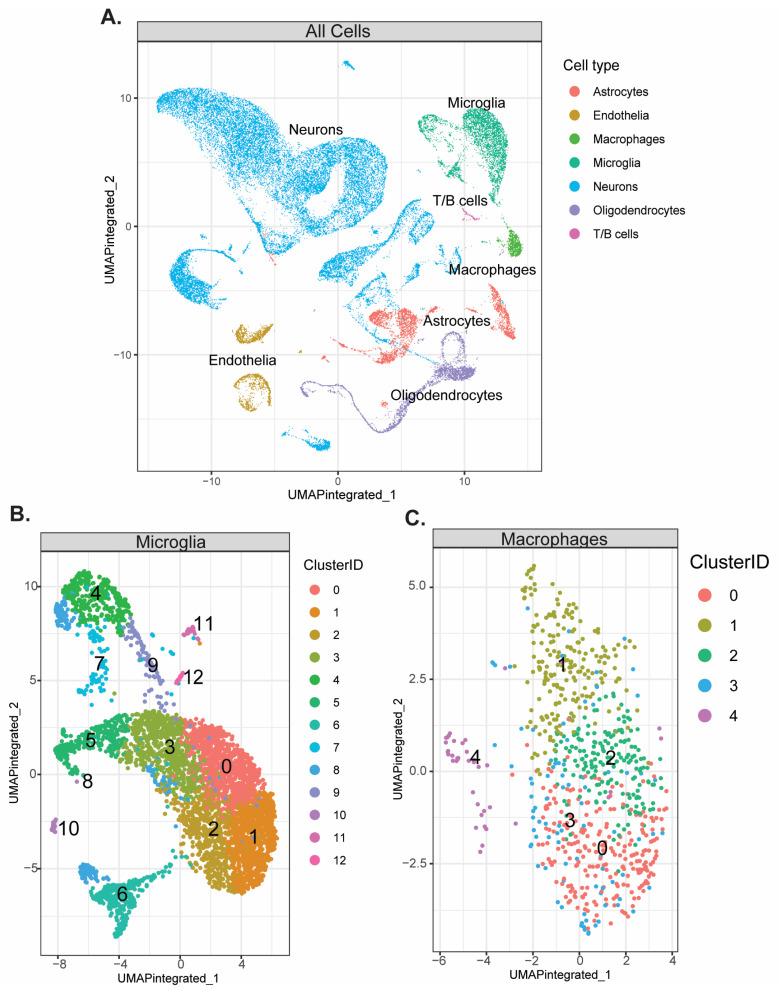
No unique subclusters emerge following HIE. ScRNAseq data from P8 and P10 combined (*n* = 6 control P8, 6 HIE P8, 6 control P10, 6 HIE P10): (**A**) Representative UMAP of all cell types identified by scRNA-Seq. (**B**) Representative UMAP of identified microglia subclusters. (**C**) Representative UMAP of identified macrophage subclusters.

**Figure 6 cells-13-01551-f006:**
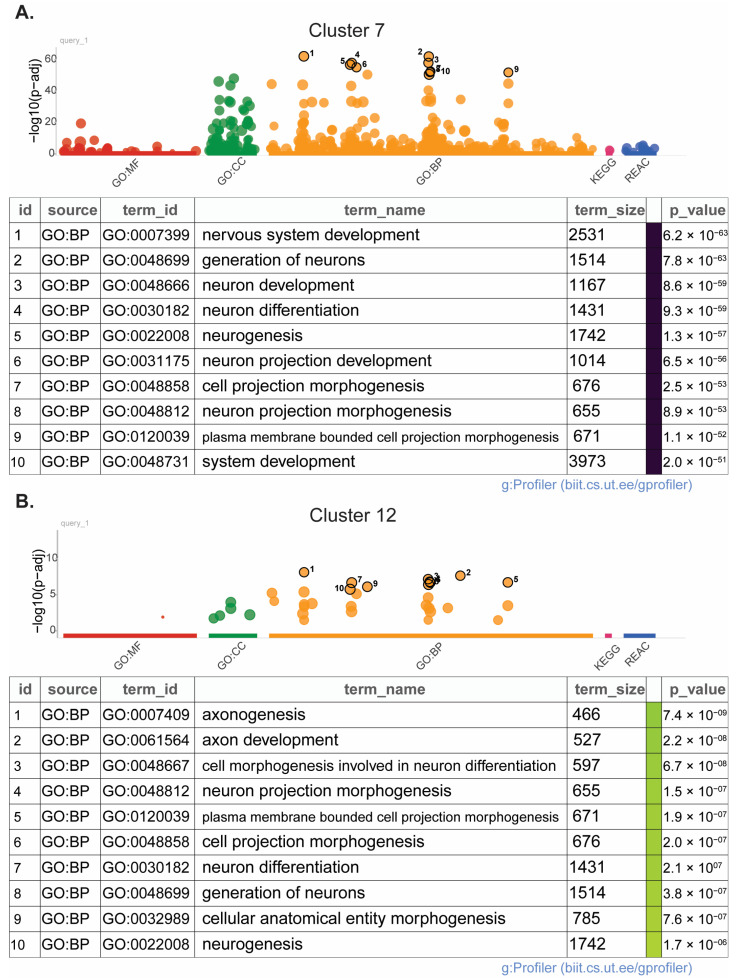
Microglia subclusters 7 and 12 emerge as clusters of interest following HIE in scRNAseq analysis: (**A**) Pathway enrichment analysis of microglia subcluster 7. (**B**) Pathway enrichment analysis of microglia subcluster 12. (*n* = 6 control P8, 6 HIE P8, 6 control P10, 6 HIE P10). (GO:BP, GOCC, GO:MF: Gene Ontology Biological Processes, Cellular Components, Molecular Functions, respectively; KEGG: KEGG PATHWAY Database; REAC: Reactome Pathway Database).

**Figure 7 cells-13-01551-f007:**
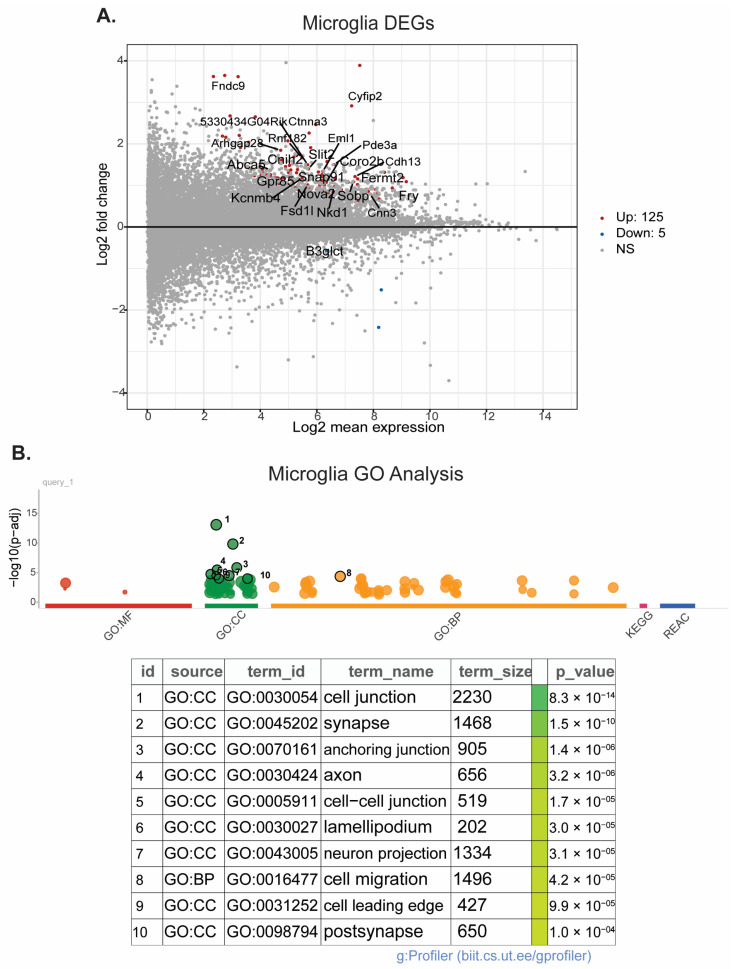
Microglia have significant transcriptional changes following HIE: (**A**) MA plot of the differentially expressed genes in microglia (P8 and P10). (**B**) Plot of the significantly different functional pathways in microglia. (*n* = 6 control P8, 6 HIE P8, 6 control P10, 6 HIE P10).

**Figure 8 cells-13-01551-f008:**
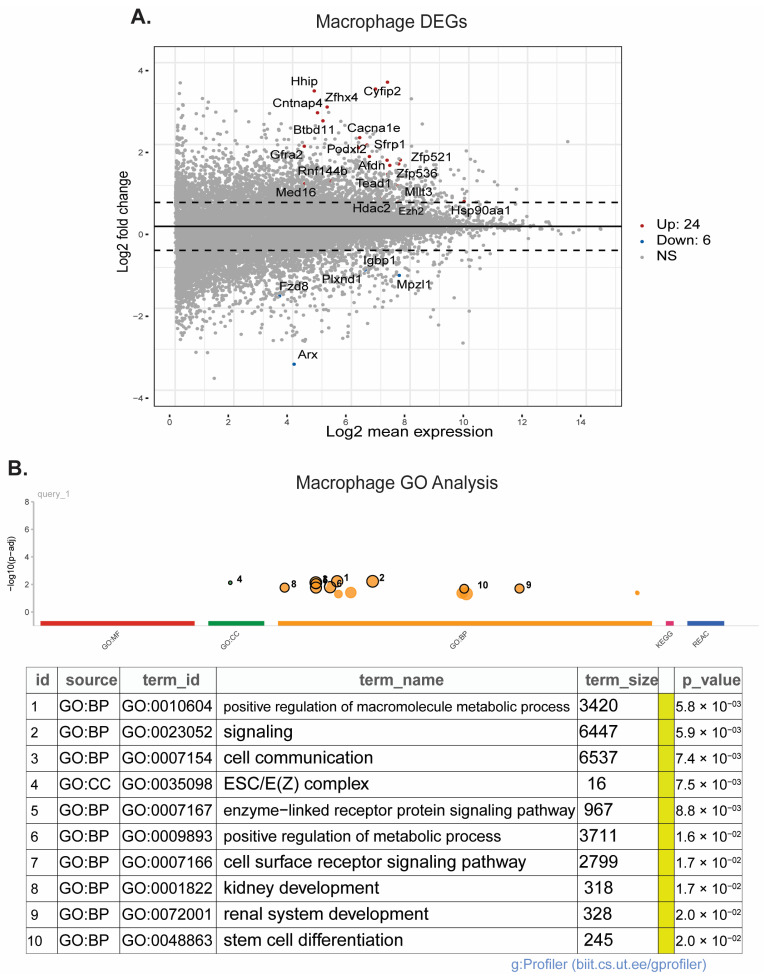
Macrophages have significant transcriptional changes following HIE: (**A**) Volcano plot of the differentially expressed genes in macrophages (P8 and P10). (**B**) Plot of the significantly different pathways in macrophages. (*n* = 6 control P8, 6 HIE P8, 6 control P10, 6 HIE P10).

**Table 1 cells-13-01551-t001:** Neonatal Acquisition of Behaviors.

Behavior	Factor	F (DFn, DFd)	*p*-Value	*p*-Value Summary
Rooting	HIE	F (1, 17) = 20.75	0.0003	***
	Sex	F (1, 17) = 0.8743	0.3629	ns
	Interaction	F (1, 17) = 0.03220	0.8597	ns
Negative Geotaxis	HIE	F (1, 17) = 5.364	0.0333	*
	Sex	F (1, 17) = 0.1637	0.6908	ns
	Interaction	F (1, 17) = 0.02355	0.8798	ns
Righting	HIE	F (1, 17) = 1.161	0.2964	ns
	Sex	F (1, 17) = 3.623 × 10^−5^	0.9953	ns
	Interaction	F (1, 17) = 0.1826	0.6745	ns
Forelimb Grasp	HIE	F (1, 17) = 0.8099	0.3807	ns
	Sex	F (1, 17) = 2.552	0.1286	ns
	Interaction	F (1, 17) = 0.3855	0.5429	ns
Hindlimb Splay	HIE	F (1, 17) = 8.031	0.0115	*
	Sex	F (1, 17) = 0.007006	0.9343	ns
	Interaction	F (1, 17) = 0.02416	0.8783	ns
Open Area	HIE	F (1, 17) = 9.012	0.008	**
	Sex	F (1, 17) = 0.01246	0.9124	ns
	Interaction	F (1, 17) = 0.003612	0.9528	ns
Air Righting	HIE	F (1, 17) = 7.354	0.0148	*
	Sex	F (1, 17) = 0.08925	0.7687	ns
	Interaction	F (1, 17) = 0.03380	0.8563	ns
Auditory Startle	HIE	F (1, 17) = 1.880	0.1881	ns
	Sex	F (1, 17) = 1.027	0.325	ns
	Interaction	F (1, 17) = 0.1742	0.6816	ns
Eye Opening	HIE	F (1, 15) = 0.5488	0.4702	ns
	Sex	F (1, 15) = 0.04480	0.8352	ns
	Interaction	F (1, 15) = 0.04480	0.8352	ns

ns = non-significant, * *p* < 0.05, ** *p* < 0.01, *** *p* < 0.001.

**Table 2 cells-13-01551-t002:** Adult Behavior Statistics.

Behavior	Factor	F (DFn, DFd)	*p*-Value	*p*-Value Summary
FP Stride Length	HIE	F (1, 40) = 8.840	0.005	**
	Sex	F (1, 40) = 1.278	0.265	ns
	Interaction	F (1, 40) = 2.384	0.1304	ns
HP Stride Length	HIE	F (1, 40) = 11.54	0.0016	**
	Sex	F (1, 40) = 1.048	0.3121	ns
	Interaction	F (1, 40) = 1.358	0.2508	ns
FP Stride Time	HIE	F (1, 40) = 3.389	0.0731	ns
	Sex	F (1, 40) = 1.261	0.2682	ns
	Interaction	F (1, 40) = 2.040	0.161	ns
HP Stride Time	HIE	F (1, 40) = 8.474	0.0059	**
	Sex	F (1, 40) = 1.359	0.2506	ns
	Interaction	F (1, 40) = 3.082	0.0868	ns
Body Speed	HIE	F (1, 40) = 0.2480	0.6212	ns
	Sex	F (1, 40) = 0.04108	0.8404	ns
	Interaction	F (1, 40) = 2.984	0.0918	ns
Grip Strength	HIE	F (1, 74) = 9.867	0.7585	ns
	Sex	F (1, 74) = 18.89	0.0024	**
	Interaction	F (1, 74) = 0.09520	<0.0001	****

ns = non-significant, ** *p* < 0.01, **** *p* < 0.0001.

**Table 3 cells-13-01551-t003:** Rotarod Statistics.

Factor	Coef	Exp (Coef)	Se (Coef)	z-Value	*p*-Value	*p*-Value Summary
HIE	−0.5373	0.5843	0.1893	−2.84	0.00455	*
Sex	0.1019	1.1073	0.1758	0.58	0.56189	ns
Interaction	0.3851	1.46.97	0.2853	1.35	0.17711	ns

coef: coefficient, exp(coef): exponential coefficient, se (coef): standard error coefficient. * *p* < 0.05

**Table 4 cells-13-01551-t004:** Microglia Upregulated GSEA Analysis.

Hallmark Gene Set	ES	NES	FDR q-Val	FWER *p*-Val	Rank at Max
TNFα Signaling via NFκB	0.58	2.88	<0.001	<0.001	2773
Allograft Rejection	0.55	2.74	<0.001	<0.001	2202
Interferon-α Response	0.60	2.71	<0.001	<0.001	4120
Interferon-γ Response	0.56	2.70	<0.001	<0.001	4099
IL6/JAK/STAT3 Signaling	0.60	2.66	<0.001	<0.001	3260
Inflammatory Response	0.47	2.39	<0.001	<0.001	1863
MYC Targets V1	0.42	2.06	<0.001	<0.001	8736
Complement	0.38	1.88	0.003	0.003	2945
E2F Targets	0.37	1.86	0.002	0.003	8678
G2M Checkpoint	0.37	1.84	0.004	0.005	8141
MYC Targets V2	0.44	1.81	0.004	0.005	8186
IL2 STAT5 Signaling	0.30	1.50	0.031	0.043	2156
PI3K AKT mTOR Signaling	0.32	1.50	0.029	0.043	5532
KRAS Signaling Up	0.30	1.45	0.035	0.057	1815
Apoptosis	0.30	1.43	0.037	0.065	2765

ES: Enrichment Score; NES: Normalized Enrichment Score; FDR: False Discovery Rate; FWR: Family-wise error Rate.

**Table 5 cells-13-01551-t005:** (**a**) Upregulated Microglia genes with subcluster and HIE interaction. (**b**) Downregulated Microglia genes with subcluster and HIE interaction.

**(a)**						
**Gene**	**baseMean**	**log2FC**	**lfcSE**	**Stat**	***p*-Value**	**Padj**
*Astn2*	30.49	2.952	1.64	44.22	1.40 × 10^−5^	1.98 × 10^−3^
*Hba-a1*	2142.28	2.279	1.73	70.31	2.80 × 10^−10^	1.86 × 10^−7^
*Hbb-bs*	6555.16	1.885	1.62	72.37	1.15 × 10^−10^	1.07 × 10^−7^
*Setbp1*	35.02	0.817	0.57	49.92	1.44 × 10^−6^	3.36 × 10^−4^
*Ptprd*	37.61	0.770	0.55	41.20	4.53 × 10^−5^	5.27 × 10^−3^
*Icam1*	69.99	0.603	0.32	49.37	1.80 × 10^−6^	4.00 × 10^−4^
*Tmtc2*	16.38	0.509	1.18	40.00	7.18 × 10^−5^	7.26 × 10^−3^
*Tuba1a*	163.35	0.429	0.62	50.20	1.29 × 10^−6^	3.15 × 10^−4^
*Hbb-bt*	751.87	0.378	1.82	52.30	5.48 × 10^−7^	1.59 × 10^04^
*Nedd4l*	45.40	0.362	0.31	44.38	1.32 × 10^−5^	1.97 × 10^−3^
*Tubb2b*	87.13	0.328	0.65	48.29	2.78 × 10^−6^	5.57 × 10^−4^
*Nfia*	199.82	0.317	0.28	55.88	1.26 × 10^−7^	5.35 × 10^−5^
*Jun*	891.49	0.188	0.29	40.09	6.94 × 10^−5^	7.18 × 10^−3^
*Rgl1*	43.30	0.186	0.21	39.80	7.76 × 10^−5^	7.68 × 10^−3^
*Maml3*	210.80	0.171	0.25	48.21	2.87 × 10^−6^	5.57 × 10^−4^
*Jund*	776.50	0.150	0.24	51.64	7.18 × 10^−7^	1.86 × 10^−4^
*Dlc1*	19.40	0.139	0.33	43.94	1.56 × 10^−5^	2.08 × 10^−3^
*Ank2*	64.41	0.134	0.33	47.34	4.08 × 10^−6^	7.59 × 10^−4^
*Klf12*	69.25	0.128	0.31	40.22	6.62 × 10^−5^	7.00 × 10^−3^
*Tmsb10*	118.80	0.105	0.37	44.61	1.20 × 10^−5^	1.87 × 10^−3^
*Rtn1*	110.34	0.093	0.46	87.06	1.83 × 10^−13^	4.25 × 10^−10^
*Nav2*	245.82	0.086	0.39	80.57	3.21 × 10^−12^	4.98 × 10^−9^
*Chd7*	72.15	0.078	0.20	45.32	9.10 × 10^−6^	1.46 × 10^−3^
*Peli2*	44.37	0.068	0.25	56.00	1.19 × 10^−7^	5.35 × 10^−5^
*Ckb*	219.35	0.065	0.17	55.52	1.46 × 10^−7^	5.67 × 10^−5^
*Sumo2*	138.34	0.048	0.14	41.08	4.75 × 10^−5^	5.39 × 10^−3^
*Dock4*	210.90	0.014	0.17	40.43	6.09 × 10^−5^	6.59 × 10^−3^
**(b)**						
**Gene**	**baseMean**	**log2FC**	**lfcSE**	**Stat**	***p*-Value**	**Padj**
*Gramd1b*	14.90	−1.315	0.49	44.12	1.45 × 10^−5^	1.99 × 10^−3^
*Kif1b*	38.76	−0.409	0.26	45.89	7.26 × 10^−6^	1.30 × 10^−3^
*Mecp2*	18.33	−0.348	0.31	45.57	8.23 × 10^−6^	1.37 × 10^−3^
*Apc*	65.68	−0.340	0.25	39.49	8.73 × 10^−5^	8.13 × 10^−3^
*Ptprs*	30.31	−0.269	0.36	54.00	2.73 × 10^−7^	9.09 × 10^−5^
*Rfx7*	21.30	−0.261	0.31	39.55	8.53 × 10^−5^	8.10 × 10^−3^
*Nav3*	414.56	−0.213	0.32	48.69	2.37 × 10^−6^	5.01 × 10^−4^
*Tcf4*	224.37	−0.200	0.25	74.41	4.76 × 10^−11^	5.54 × 10^−8^
*Ttc3*	63.26	−0.200	0.27	42.15	3.14 × 10^−5^	3.75 × 10^−3^
*Ppp3ca*	133.77	−0.199	0.15	44.22	1.40 × 10^−5^	1.98 × 10^−3^
*Tnik*	26.99	−0.149	0.94	45.56	8.27 × 10^−6^	1.37 × 10^−3^
*Spag9*	84.88	−0.131	0.20	54.55	2.18 × 10^−7^	7.81 × 10^−5^
*Pld1*	35.21	−0.121	0.20	40.65	5.60 × 10^−5^	6.20 × 10^−3^
*Arsb*	446.82	−0.104	0.12	39.69	8.10 × 10^−5^	7.85 × 10^−3^
*Meis1*	30.00	−0.097	0.54	55.87	1.26 × 10^−7^	5.35 × 10^−5^
*Basp1*	376.98	−0.094	0.14	129.34	8.39 × 10^−22^	3.90 × 10^−18^
*Ssh2*	174.40	−0.091	0.19	42.30	2.96 × 10^−5^	3.63 × 10^−3^
*Ddah2*	51.16	−0.066	0.28	42.97	2.29 × 10^−5^	2.96 × 10^−3^
*Celf2*	241.00	−0.064	0.39	71.32	1.82 × 10^−10^	1.41 × 10^−7^
*Hsp90ab1*	360.71	−0.060	0.10	68.87	5.20 × 10^−10^	3.02 × 10^−7^
*Zbtb20*	141.23	−0.057	0.28	53.38	3.52 × 10^−7^	1.09 × 10^−4^
*Marcks*	664.14	−0.052	0.09	42.31	2.95 × 10^−5^	3.63 × 10^−3^
*Fosb*	120.36	−0.033	0.31	52.02	6.16 × 10^−7^	1.69 × 10^−4^

## Data Availability

RNA-seq data are available in NCBI Geo, accession number GSE275713. All other data are available upon request.

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
