# Peer review of "A Novel Non-Invasive Murine Model of Neonatal Hypoxic-Ischemic Encephalopathy Demonstrates Developmental Delay and Motor Deficits with Activation of Inflammatory Pathways in Monocytes"

_cells, 2024, doi:10.3390/cells13181551_

Round 1
Reviewer 1 Report
Comments and Suggestions for Authors
The Authors want to introduce a novel model of neonatal hypoxic/ischemic encephalopathy based on a non-invasive two-hit strategy, first the administration of proinflammatory agents in the pregnant mother, and then of a short but severe hypoxia bout at postnatal day 6. The Authors tested HIE pups by behavioral and RNA evaluations using up-to-date single-cell analysis, that highlights potentially interesting phenotypes. The numerosity of the tested populations seems appropriate, but several inconsistencies must be solved to promote a preliminary study into a full one.
It is not evident to me why the hypoxia bout was administered at P6 rather than earlier to better approximate human HIE that usually occurs at birth.
Figure 1 covers the operations done until P6, but it must be extended to include the rest of the experimental design and the group assignment. Clearly identify using the Px code in Figure 1, in the results, and in the respective figure legends at which time after birth the various measurements were taken, along with the numerosity in the various subgroups.
MIA data might be considered more valuable than untreated pups as negative controls as they represent a one-hit model. However, only a small part of the reported measurements refers to MIA, and RNA analysis does not consider MIA at all. In line 426 it is said “Future studies can further elucidate the individual and combined effects of MIA and hypoxia within this model”. Although showing both MIA and HIE data here would lead to a complete study, the Authors must decide whether showing MIA data at all in this manuscript. Perhaps the outline of the manuscript would be weakened but the clarity of presentation could benefit.
The unexpected outcome of the male vs. female comparison is a weak point that deserves more results and discussion. Please report sex-dependent comparisons also for other data and specifically for RNA analyses. It is not clear to me if the RNA analyses refer to males or females.
HIF signaling may be quite relevant here, please comment.
The meaning of the dashed lines in Figure 2 is not clear.
Progressive hypoxia-anoxia was necessarily followed by reoxygenation, meaning that the observed damage may have stemmed not from hypoxia-ischemia but rather from the reoxygenation-reperfusion of hypoxic-ischemic tissue. Please comment.
Add a reference to ethical clearances.
Add anesthesia details.
Specify the mortality rate of the procedure.
Lines 102 and 126 and 402, cite the references correctly.
Line 323, female missing.
Comments on the Quality of English LanguageThe language is sometimes difficult to follow. Non-English-speaking readers may benefit from minor editing of complex sentences.
Reviewer 2 Report
Comments and Suggestions for Authors
The current manuscript is entitled "A novel noninvasive murine model of neonatal hypoxic ischemic encephalopathy demonstrates developmental delay and motor deficits with activation of inflammatory pathways in monocytes" and is proposed by Wright-Jin et al. The study proposes an interesting murine model for behavioural impairments seen in HIE. The group followed the two-hit hypothesis by generating MIA with LPS and hypoxic shock to the 6-day pups.
Several recommendations:
- Statistics, line 251: it would be beneficial to mention what non-parametric tests were performed; also, the Authors could explain why they are stating that t-test was performed when the Figure legends say that one-way ANOVA was;
- Lines 267-277: the information could better be fitted for the Animals and Methods section;
- Figures 2-6: the charts are small and this could constitute a disadvantage in understanding this outstanding data; also, the text in the tables included in the figures seems non-readable; this could be improved by increasing the figures and the font sizes;
- Figure 2: the Authors should mention in the legend the meaning of the statistical significance symbols they used (*, **, and ***);
- Line 402: there might be a misplaced link in the text;
- The definitions of abbreviations should only be mentioned once when first used; the Authors could kindly revise this aspect as abbreviations are repeatedly defined throughout the manuscript;
- The conclusions section is missing; the Authors could kindly formulate the conclusion separately while taking into account the discussions, the advantages and the limitations of the proposed model.
Best wishes.
Round 2
Reviewer 1 Report
Comments and Suggestions for Authors
To help Reviewers, the Authors should supply a version of the manuscript marked for corrections. For example, this Reviewer can’t locate the valuable Authors’ observation regarding the term equivalence in mice, which should have been added to line 108 (first issue of the reply).
MIA data. I suggest removing MIA data from Fig 2A and adding a short text note on the maintained whole brain volume as “Data not shown”.
Male vs. female. I understand the concept, although it is somewhat difficult to grasp as it is not linear reasoning. Although the concern may be considered answered, I suspect that readers may enjoy help with an additional figure in Supplemental data.
Dashed line in Fig 2. Stating “The dashed line indicates the day of hypoxia exposure (P6)” should be replaced by something like “The dashed line indicates the situation at the day of hypoxia exposure (P6)” or so. Consider replacing the line with a band representing the 95% confidence limits or mean/SD.
Reoxygenation-reperfusion. I am glad that the Authors recognize this as a critical issue, but this must be highlighted as an unavoidable (?) limit to their study that may complicate the interpretation of the data.
Ethical clearance. I am asking the Editorial Board to clarify this issue and hope they will contact the Authors directly.
All other concerns have been solved.
